# What Is Hidden behind Amputation? Quanti-Qualitative Systematic Review on Psychological Adjustment and Quality of Life in Lower Limb Amputees for Non-Traumatic Reasons

**DOI:** 10.3390/healthcare11111661

**Published:** 2023-06-05

**Authors:** Laura Calabrese, Marina Maffoni, Valeria Torlaschi, Antonia Pierobon

**Affiliations:** Istituti Clinici Scientifici Maugeri IRCCS, Psychology Unit of Montescano Institute, 27040 Montescano, Italy

**Keywords:** amputation, amputee, lower limb, psychological adjustment, well-being, health-related quality of life, anxiety, depression

## Abstract

Objective: This systematic review aims to investigate Quality of Life (QoL)/Health Related Quality of Life (HRQoL) and psychological adjustment in non-traumatic lower limb amputees (LLA). Methods: PubMed, Scopus, and Web of Science databases were used for the literature search. Studies were read and analysed using the (Preferred Reporting Items for Systematic Reviews and Meta-Analyses) PRISMA statement procedure. Results: The literature search retrieved 1268 studies, of which 52 were included in the systematic review. Overall, psychological adjustment, especially depression with or without anxiety symptoms, influences the QoL/HRQoL in this clinical population. Other factors influencing QoL/HRQoL include subjective characteristics, physical aspects, the cause and level of the amputation, relational aspects, social support, and the doctor-patient relationship. In addition, the patient’s emotional-motivational status, depression and/or anxiety symptoms, and acceptance play a key role in the subsequent rehabilitation process. Conclusions: In LLA patients, psychological adjustment is a complex and multifaceted process, and QoL/HRQoL may be influenced by various factors. Shedding light on these issues may provide useful suggestions for promoting clinical and rehabilitative interventions that may be tailored and effective in this clinical population.

## 1. Introduction

Amputation is the surgical removal, or accidental loss, that involves the elimination of part or all of a limb [1]. It therefore refers to an acquired condition caused by injury, disease, or surgery [2]. This procedure can be used when arterial reconstruction surgery is not technically possible, has failed, or when the limb has lost its function. For example, amputation can be the result of various conditions such as peripheral vascular disease, trauma, malignancy, metabolic disease, and infection [2]. According to the literature, lower limb amputation (LLA) represents 80–85% of all amputations and is mainly due to vascular diseases such as diabetes mellitus, atherosclerosis, and Buerger’s disease [3].

The choice of amputation level is based on the patient’s postoperative function and the best possible primary wound healing [4]. In this sense, the concept of lower limb amputation can encompass different types of amputation, allowing different issues and treatments to be considered in order to provide comprehensive patient care.

### 1.1. Physical and Psychological Well-Being

The literature and clinical experience show that amputation has a significant impact on physical and psychological well-being, resulting in various life changes [5]. In fact, one of the most important consequences is the functional limitation, which, if not addressed, may lead to permanent disability, and, in turn, may be experienced as a traumatising experience [6]. The loss of a limb is therefore a significant experience that causes disruption in several areas of the person’s life: change in mobility [7], participation in social activities [7], possibility of returning to work [8], and the patient’s mood [9].

In addition, amputation has an impact on psychological well-being, vulnerability, and the definition and redefinition of identity [7]. The loss of a limb also affects the perception of body image. Specifically, increasing levels of amputation are associated with greater discomfort with body image [5]. In addition, amputation is associated with anxiety, depression, social isolation, and pain [10]; it even predicts a change in leisure activities [11] and social position within the community [7]. 

Overall, the loss of a body part may lead to a period of grief that requires time for readjustment. During this period, physical, functional, and psychological problems related to the difficulty of adjustment may occur [7]. 

In summary, there are several factors that influence the outcome of this period, and various studies have focused on physical factors, especially the age of the amputee [12], the level of amputation, and the presence of comorbidities [13].

### 1.2. Rehabilitation after Amputation

After amputation, patients begin a rehabilitation process that includes adapting to various physical and psychosocial challenges [14].

Factors that have a positive impact on rehabilitation include the ability to perform activities prior to amputation, no or little delay in admission to a rehabilitation centre, patient motivation, and good communication with the rehabilitation team. In this regard, the presence of a multidisciplinary team appears to promote a positive outcome for the amputee [15]. The team, working in a coordinated manner, aims to bring the amputee to the highest level of functional recovery in relation to his or her potential [7]. This can be achieved through the use of prostheses, which counteract the negative effects of amputation by helping to reduce comorbidity [16]. Prosthetics can be seen as a relevant indicator of quality of care and quality of life. Patient satisfaction plays a key role in the recovery of mobility and adherence to treatment [7]. However, it should be emphasised that more than 40–60% of amputees are not satisfied with their prostheses. Failure to overcome the initial problems caused by the amputation difficulties in fitting and wearing the prosthesis may lead to sporadic use of it or even complete rejection [13,15].

### 1.3. Quality of Life and Psychological Adjustment

Amputation is therefore an intervention that affects the whole of the amputee’s existence and, in turn, his or her Quality of Life in general terms (QoL) and Health-Related Quality of Life (HRQoL) in particular [17].

Specifically in amputees, QoL/HRQoL is associated with several aspects such as prosthesis use [7], presence of pain [18], occupational reintegration [18], level of amputation [18], and social support [19]. Studying QoL/HRQoL in amputees remains complex, hampered by methodological issues such as heterogeneity concerning samples and measures [20].

Psychological adjustment is the process activated in response to illness, a disabling event, and its treatment [21], and is a key factor to consider in amputees. Individuals must come to terms with the new physical condition, the various psychological consequences, and the underlying causes of the amputation [22]. There is therefore a large interindividual variability in response to the same adverse event, specifically amputation. For example, different reactions may depend on the patient’s gender, age, personality factors, social support, experience of pain, cause of amputation, and time since surgery [23]. However, there are some common reactions, which include anxiety, reduced quality of life, depressed mood, pain, body image concerns, dysfunctional coping strategies, and difficulties in social interactions [13,24]. Amputees therefore experience a range of complex psychological reactions. Emotional difficulties may interfere with functional recovery and rehabilitation itself. Among these, depression remains the most common response, negatively affecting psychological and physical adjustment [1]. The presence of postoperative depressive symptoms is common [25], but these appear to decrease over time [26]. Anxiety is also associated with the fear of losing physical function and mobility. Another aspect that influences the outcome of psychological adjustment is the coping strategy adopted. Coping refers to the way in which adverse situations are managed and the responses that are used to deal with them. Specifically, people who tend to focus on a problem-solving approach are associated with a positive psychosocial adjustment outcome [27] and increased functionality; conversely, this coping style seems to be negatively associated with depression [22]. 

Thus, considering how many aspects may influence the well-being of LLA patients, this systematic review aims to provide an overview of the current state of the literature on quality of life and psychological adjustment in non-traumatic LLA patients, highlighting risk and protective factors that have not yet been studied in depth.

## 2. Materials and Methods

A systematic review was conducted to identify articles related to psychological adjustment and QoL/HRQoL in lower limb amputees. Data were analysed and reported according to the international PRISMA (Preferred Reporting Items for Systematic Reviews and Meta-Analyses) guidelines [28].

### 2.1. Search Strategy and Data Extraction

Three publicly accessible databases (PubMed, Scopus, and Web of Science) were used for the electronic literature search using the following terms: (“lower extremity amput*” OR “lower limb amput*”) AND (psychological OR acceptance OR adherence OR compliance OR “health-related quality of life”). Articles published between 2001 and 2021 were included in the search. We combined Boolean operators and wildcard characters appropriately to focus the search and detect plural and singular forms of the same terms in all databases. We also included synonyms or spelling variations. As the index terms varied between databases, the choice of terms was checked both by clinicians with specific expertise in amputation and by reading sentinel articles.

Two reviewers (L.C. and M.M.) separately screened the retrieved records after the completion of the electronic search, starting with titles considered potentially relevant. The reading of the abstracts filtered out the records considered eligible. Papers without abstracts were immediately disqualified as they could not be screened properly. Finally, the full text was screened to find papers relevant to the scope of the review.

A third reviewer (A.P.) helped to resolve inconsistencies in the inclusion and exclusion criteria between L.C. and M.M., and the papers went through the selection process with the full agreement of the authors. In addition, the papers that were finally considered eligible were also discussed with the other author (V.T.) to check their consistency with the aim of this review.

In terms of data extraction, the authors collectively decided what information might be relevant based on the focus of the review, clinical experience, and previously published reviews. Therefore, a table was created to highlight the relevant data. Specifically, L.C. and M.M. separately retrieved the data from each article to complete the table. Then, A.P. and V.T. discussed any inconsistencies. Finally, the authors re-read the full text of each article to ensure the accuracy of the relevant data reported in the tables.

### 2.2. Inclusion and Exclusion Criteria

Articles were considered eligible if they were written in English and published in peer-reviewed journals. Both qualitative and quantitative research was considered, in particular cross-sectional, longitudinal, and intervention studies. More specifically, articles were included in the systematic review if they reported the perspective of adult patients with a lower limb amputation due to a clinical condition (e.g., diabetes or vascular disease). Publications were included if they discussed amputation in relation to psychological aspects (e.g., QoL/HRQoL, anxiety, depression, coping). 

Grey literature and articles dealing with the validation of scales and questionnaires were excluded. Studies involving patients of developmental age (<18 years) and amputations due to traumatic events (car accidents or war) were also excluded. Articles that included upper limb amputation were also excluded from the current systematic review. We also excluded articles that were drug trials or that focused on medical and technical issues. Publications that only considered the carer’s or healthcare professional’s point of view were not included.

## 3. Results

After searching the databases and removing duplicates, 720 records were found. The titles and abstracts were screened, and 256 eligible articles were found. After full-text screening, 52 publications were eligible for inclusion in this systematic review. The main reasons for exclusion were that many studies did not focus on psychological constructs (n = 102) or reported a traumatic event as the cause of amputation (n = 96) (Figure 1). 

The total number of amputee patients included in this work was 5529, and the sample sizes ranged from studies including 6 to 821 patients (age range: from 36 to 90 years, the majority of whom were aged over 60 years). Not all studies reported whether the sample used prostheses or other walking aids (n° 30, 57.7%). The main cause of amputation was diabetes (n° 28), followed by vascular reasons (n° 19). It should be noted that several articles included samples with different organic causes of amputation.

Table 1 summarises the results in terms of the countries of origin of the studies. In short, most of the studies were conducted in Europe, specifically in the United Kingdom (15.4%) and Portugal (15.4%). More than half of the articles had a quantitative design (75%), the others had a qualitative design (21.2%), and the rest were psychological interventions (3.8%). 

The articles showed that QoL/HRQoL, and psychological adjustment are based on a variety of constructs. It should be noted that since some articles focused on patients’ perceptions of quality of life in general (QoL) and others on quality of life in relation to health status (HRQoL), we decided to keep both acronyms in the results and discussion sections.

Several articles addressed QoL/HRQoL, depression, anxiety, social support, and prosthesis use (Table 2). These topics were reported in both quantitative and qualitative articles. The qualitative articles focused mainly on the subjective experience of being a lower-limb amputee. Focus groups or interviews (either face-to-face or by telephone) were used to explore the constructs of interest. In quantitative studies, there was wide variability in the instruments chosen. The most commonly used instruments were HADS (Hospital Anxiety and Depression Scale) (n° 13), SF-36 (n° 11), and WHOQOL-BREF (World Health Organisation Quality of Life Brief Version) (n° 9). Regarding the functional index, the Barthel scale is the most commonly used and is difficult to relate to a single construct but rather to a broader measure of the patient’s functionality [22,29,30,31,32]. Both quantitative (75%) and qualitative (21.2%) studies highlighted factors linked to QoL/HRQoL. For this reason, we considered QoL/HRQoL to be the core construct of our review (Figure 2).

To provide a clearer understanding, we decided to analyse the results based on the design used in the studies.

### 3.1. Findings from Quantitative Research

In the quantitative articles, a relevant issue is the decrease in QoL/HRQoL soon after the amputation [19,27,42,58,61,64,65]. However, QoL/HRQoL tended to improve over time, especially after 6 months [55,64] or after 1 year [29]. In addition, studies showed a negative association between QoL/HRQoL and anxiety [22,23,26,27,32,44,45,46,48,57] and depression [10,22,23,26,27,32,42,44,45,46,48,57,68]. In particular, studies by Pedras et al. [46] reported that increased preoperative anxiety and postoperative depression were associated with decreased QoL/HRQoL. 

The results showed that factors influencing QoL/HRQoL improvement were social support [11,19,22,23,27,46,47] and prosthesis use [19,27,41,49,55,58,59,60]. The latter was particularly dependent on fear of falling [30,31,55], pain [34], emotional and psychological factors [49], such as coping strategies used [27,49], anxiety [49], and depression [49]. 

Another aspect relevant to QoL/HRQoL in amputees was the level at which the amputation was performed; transtibial amputations correspond to a higher level of health than transfemoral amputations [11,27,63,65,66,67]. Other relevant factors for QoL/HRQoL were: time since amputation [57], persistence of pain [19,27,47,48,57,65], body perception image [27,42,53], coping strategies [22,27,51], age [19,57,58], gender [11,63], inability to walk [57], and acceptance of own illness and clinical conditions [35]. In this context, disease acceptance implied an increase in functional independence and an overall improvement in QoL/HRQoL [35,53]. It has also been described that disease acceptance is lower in subjects who still experience pain [36].

Appendix A (Table A1) shows the quantitative articles included in the current systematic review.

### 3.2. Findings from Qualitative Research

From a qualitative point of view, social support [39,54] and emotional support [33] seem to play an important role. Other findings highlighted the amputation experience in general, from the time when the person made the decision to undergo amputation to the experience of regaining partial independence [50]. During this journey, amputees may experience a loss of control over their lives, both at the level of social relationships and their role in society and at the physical level (i.e., mobility and functionality) [6,54]. The loss of mobility due to amputation can lead to isolation, which in turn threatens future expectations [37,54]. The study by Torbjörnsson et al. [38] showed that some people perceived greater benefits (e.g., reduced pain, reduced risk of death) than the costs of the amputation.

Appendix B (Table A2) shows the qualitative articles included in this systematic review.

### 3.3. Findings from Psychological Intervention Studies

Two intervention studies were included in this systematic review. These looked at the rehabilitation period, which focused mainly on occupational/physiotherapy interventions [56] and desensitisation [40]. The first included a change in QoL/HRQoL and functional independence, and the second led to a reduction in pain levels.

Appendix C (Table A3) shows the psychological intervention studies included in this systematic review.

## 4. Discussion

This systematic review aims to explore QoL/HRQoL and the psychological adjustment of non-traumatic lower limb amputees. Overall, the results led to the identification of several factors related to this health condition. 

From a descriptive point of view, half of the studies were conducted in Anglo-Saxon countries (UK, USA, Canada) and Portugal, suggesting a specific interest in studying QoL/HRQoL and psychological manifestations in people with disabilities or physical limitations. 

In terms of conditions leading to amputation, diabetes and vascular problems were the most common, due to the progressive worsening of clinical conditions associated with these diseases and poor long-term adherence. The majority of articles on LLA focused on psychological adjustment, adherence/compliance and QoL/HRQoL assessment; only two articles investigated educational or psychological interventions. In addition, most studies, both quantitative and qualitative, described QoL/HRQoL as a core construct.

### 4.1. QoL/HRQoL over Time

Overall, QoL/HRQoL was significantly lower in LLA, as reported in previous reviews [69,70]. Nevertheless, results also showed that at some point after amputation (i.e., 6 months to 1 year after surgery), QoL/HRQoL improved [29,55,64]. This improvement has been described as depending on several factors, such as the enhancement of physical performance in terms of functionality and mobility, the emotional and motivational aspects, the social support received, and the amputee’s characteristics. In the literature, there is still disagreement as to which characteristics may play a more relevant role in patients’ psychological adjustment and their QoL/HRQoL: the disease that led to the amputation [26], age [19,57,58], gender [11,63], inability to walk [57], body image [27,42,42,53]. For example, amputation level has been described in previous literature as a key factor influencing QoL/HRQoL. Specifically, higher amputation levels, such as transfemoral, correspond to lower QoL/HRQoL than transtibial amputation [63,65,67,71]. These data may be explained by the fact that transfemoral amputations may require longer periods of rehabilitation to regain function and independence than trans-tibial amputations. 

In this context, another relevant aspect that emerged from this review is the use of prostheses and the ability to walk again. It is well known that prostheses are a means of regaining independence and motor function and therefore have an overall impact on QoL/HRQoL. In this regard, Davie-Smith et al. [69] found that the ability to walk influenced participation in social activities and the ability to live independently. From a daily clinical perspective, prosthesis use depends on comorbid conditions, social functioning, amputation level, and the patient’s motivation. In our review, the factors that negatively influenced prosthesis use were the presence of pain [34], the risk of falling [30,31,55], anxiety [49], and depression [49]. Coping behaviours may play a positive or negative role too, depending on the strategies used [27,49]. These factors were also found by Luza et al. [72], who showed that physical adaptation may also depend on age, education level, and daily use of the prosthesis. It is important to note that not all articles included in our review specified whether amputees used a prosthesis or other walking systems. According to us, this lack of detail is a gap that further research should fill in order to better understand patients’ needs and possible difficulties and resistance towards specific types of devices or interventions proposed to them.

### 4.2. QoL/HRQoL and Psychological Constructs

As shown in Table 2, most studies have investigated and confirmed the role of depression and anxiety in this clinical population. These data are consistent with what is known about amputees, regardless of the reason for surgery. Indeed, in a recent review by Sahu et al. [73], symptoms of anxiety and depression were present and improved over time in lower-limb traumatic amputees. Anxiety and depressive symptoms are negatively correlated with QoL/HRQoL, highlighting how these constructs are risk factors for the success of the rehabilitation process [23]. Thus, mood dysregulation has an impact on motivation and negatively affects adaptation. It should be noted that higher levels of depression have been found in this population than in other hospitalised patients [26]. In addition, Pedras’ studies [22,45,46] have shown that the presence of both increased preoperative anxiety and postoperative depression correlates with a decrease in QoL/HRQoL. Thus, anxiety and depressive symptoms deserve special attention as they may affect rehabilitative outcomes and pose barriers to patients’ motivation and engagement. 

An interesting aspect that emerged from the articles concerns knowledge about the amputation process [13,17]. Amputees in Torbjörnsson’s group [38] reported that they did not feel involved in the amputation decision and had not received enough information, which made the acceptance process difficult. In fact, amputation is a complex and personal experience that involves regaining independence and accepting the loss of a body part [34]. The amputee is therefore required to reconfigure her/his role at work, in the family, and in society [37,54]. Acceptance is thus a multifaceted process of life reorientation with respect to the new condition [6]. Indeed, acceptance of the illness involves coping with difficult experiences that may include grief, anxiety, and embarrassment [74]. The ability to cope with these internal experiences has an impact on maintaining high QoL/HRQoL [35]. For example, amputees who still reported pain were found to have lower disease acceptance [36]. Subjects who received more benefits from the amputation (such as less pain and a reduced risk of death) reported greater satisfaction and acceptance [17,35,38,53]. 

Coping strategies also play a key role in the acceptance process. Indeed, the amputation experience involves a loss of control over one’s life; therefore, the amputee tries to reorganise himself with respect to the new situation by using different coping strategies [6]. Coping strategies appear to contribute to a person’s psychological well-being and reduce the negative impact of amputation [27]. QoL/HRQoL is influenced by problem-focused strategies [27,50] and motivation to regain independence [17]. Therefore, it is recommended to propose and investigate psychoeducational interventions aimed at promoting and fostering appropriate problem-solving and coping skills in dealing with the new challenges of daily living.

Furthermore, body image, which is still poorly investigated in non-traumatic LLA, was found to be a predictive component of QoL/HRQoL [27]. Body image negatively correlates with QoL/HRQoL [27] and has an impact on psychosocial outcomes [43]. It must be said that the few results on body perception and body image suggest the need for further research to pay more attention to the body component in this clinical population. The body without a limb part needs a new tool with which the patient must explore the world and start the process of acceptance. Thus, the patient must find a new psychophysical balance and a satisfactory QoL/HRQoL.

Social support can be considered a protective factor and is associated with better mental health [19]. It mitigates negative outcomes and has been positively correlated with resilience—the ability to cope with and overcome a traumatic event [42]. Social support refers not only to family and friends but also to other amputee patients. In particular, peer support allowed them to compare with each other, create clearer and more realistic expectations about the amputation [13], and help define practical and achievable goals, helping to accept the changed situation [38]. These findings suggest the importance of a global approach to patient care, including attention to the social network and carers who can help the amputee cope with the new health status. Again, findings highlight the importance of a biopsychosocial approach that focuses not only on medical and rehabilitation needs but also on psychological and social needs [75].

Finally, only a few articles reported on interventions in this clinical population. These studies focused on rehabilitation time, with an emphasis on the variation in QoL/HRQoL following multidisciplinary treatments [56] and the reduction of pain through tactile desensitisation [40,76]. Further research is needed to better understand the role of a multidisciplinary approach involving different healthcare professionals (doctor, nurse, physiotherapist, psychologist, and dietician). It is important to understand how rehabilitation interventions could contribute to positive psychological adjustment, taking into account the relevant process of constant self-redefinition throughout life [77].

### 4.3. Future Research, Strengths, and Limitations

Overall, this systematic review is one of the few attempts to synthesise the evidence on various psychological aspects of non-traumatic lower limb amputation. To our knowledge, it is the only review that considers the relationship between QoL/HRQoL and psychological adjustment from a multifaceted perspective, drawing on evidence and suggestions from quantitative, qualitative, and interventional studies. The findings may pave the way for future research and interventions tailored to this clinical population. For example, multidisciplinary programmes are more than welcome in order to promote greater awareness among patients of what this type of surgical intervention will mean for them from a functional, psychological, and social point of view. Making patients more informed, motivated, and strategic can indeed improve adherence to medical and behavioural treatments and thereby improve outcomes, as suggested by the Three Factor Model [78]. These results can be achieved through group classes that provide patients with knowledge on medical, functional, and nutritional aspects, as well as psychoeducation tips to promote the best possible QoL/HRQoL and counteract depression or anxiety. Future research is therefore recommended to test which combinations of interventions may be more effective in maximising the well-being of these patients.

Despite possible merits, there are some limits that need to be discussed. Firstly, there was no quality assessment of the articles, which may lead to the inclusion of poor-quality research or studies with different characteristics. For example, it should be noted that not all articles specified whether participants used prostheses or other walking aids, which may have influenced the relevance given to some factors. This choice was made with the aim of collecting all data on the topic in order to provide suggestions and tips for clinical practise and to suggest well-structured observational and interventional studies. Secondly, no meta-analysis has been performed. Further research is recommended to fill this gap. Thirdly, the comparison of research studies may be challenging due to the differences between healthcare systems and cultures around the world regarding lower limb amputees and rehabilitation interventions. Thus, the results may be biased by the lack of certain cultural considerations that should have explained the decision to investigate only some aspects and neglect others. Fourthly, the choice of search terms may have introduced a further bias, as may the lack of comparison between traumatic and non-traumatic amputations. Finally, there were no differences in the clinical conditions leading to amputation, which may be another aspect to investigate in future research. 

## 5. Conclusions

This review shows that amputation is a complex process, not just a physical event. QoL/HRQoL in lower limb amputees is initially lower but can be improved through various factors: the level of amputation and the medical condition that caused it; general clinical conditions; perceived social support; motivation; individual and social characteristics; and the presence of depression, anxiety, and coping strategies. In addition, QoL/HRQoL may be influenced by the whole rehabilitation process, which in turn may influence amputees’ QoL/HRQoL. The findings highlight the need to develop multidisciplinary interventions that address not only the physical aspect but also the psychological and social dimensions to improve QoL/HRQoL. Future studies could examine changes identified in this review at different stages of the amputation process, particularly from the decision to amputate to physical rehabilitation and then to prosthesis fitting. Overall, according to the authors, these findings provide useful suggestions not only for research but also clinicians: positive psychological adjustment to the daily challenges can promote better adherence, QoL/HRQoL and medical outcomes in amputees.

## Figures and Tables

**Figure 1 healthcare-11-01661-f001:**
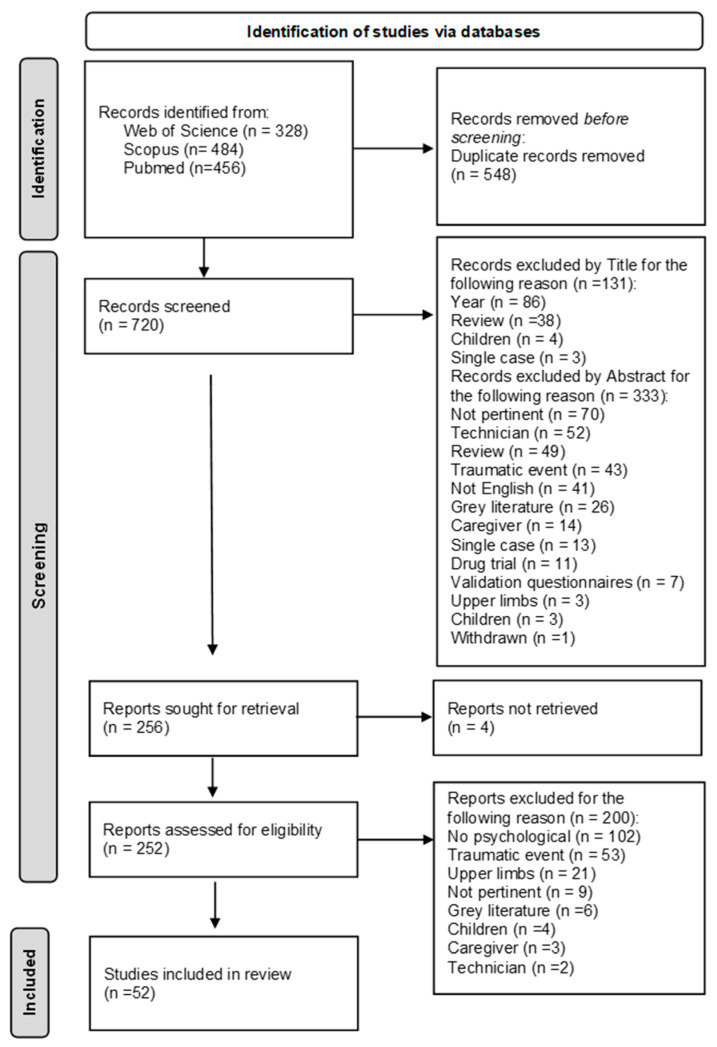
PRISMA flowchart. Represents the systematic review selection process by indicating the number of excluded articles and the reasons.

**Figure 2 healthcare-11-01661-f002:**
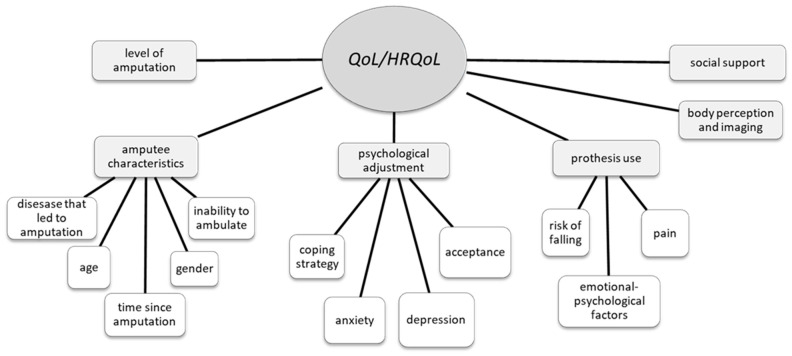
A visual map of factors associated with QoL/HRQoL in LLA for non-traumatic reasons.

**Table 1 healthcare-11-01661-t001:** Results related to nations.

Nation	HD (Ranking)	n (%) 52
UK	0.932 (13)	8 (15.4%)
Portugal	0.864 (38)	8 (15.4%)
Canada	0.929 (16)	6 (11.5%)
USA	0.926 (17)	4 (7.7%)
Sweden	0.945 (7)	3 (5.8%)
Australia	0.944 (8)	2 (3.8%)
Ireland	0.955 (2)	2 (3.8%)
Netherlands	0.944 (8)	2 (3.8%)
Turkey	0.820 (54)	2 (3.8%)
Poland	0.880 (35)	2 (3.8%)
Serbia	0.806 (64)	1 (1.9%)
Europe	-	1 (1.9%)
France	0.901 (26)	1 (1.9%)
Germany	0.947 (6)	1 (1.9%)
Hungary	0.854 (40)	1 (1.9%)
Jamaica	0.734 (101)	1 (1.9%)
Malaysia	0.810 (62)	1 (1.9%)
Pakistan	0.557 (154)	1 (1.9%)
Romania	0.828 (49)	1 (1.9%)
Saudi Arabia	0.854 (40)	1 (1.9%)
Sudan	0.510 (170)	1 (1.9%)
Taiwan (China)	0.761 (85)	1 (1.9%)
Trinidad and Tobago	0.796 (67)	1 (1.9%)

The HDI (Human Development Index) is based on three dimensions: (a) life expectance at birth; (b) expected years of schooling and mean years of schooling; and (c) gross national income per capita (United Nations Development Programme, http://hdr.undp.org/en, accessed on 20 July 2022).

**Table 2 healthcare-11-01661-t002:** Main constructs, in alphabetical order, emerging from the review and related references.

Constructs	Definitions	References
Acceptance	The subject’s awareness of the new physical condition	[1,6,10,13,17,33,34,35,36,37,38,39]
Anxiety	Psychological state characterised by worry and apprehension	[[22] ^d^] [[23], ^d^] [[26], ^d^] [27] [[32], ^d^] [[36], ^d^] [[40], ^d^] [[41], ^d^] [[42], ^d^] [[43], ^d^] [[44], ^d^] [[45], ^d^] [[46], ^d^] [[47], ^d^] [48]
Coping strategies	Behaviours used to manage, minimise, and control stressful or negative events	[13,17,22,23,27,37,39,47,49,50,51,52]
Depression	Psychological state characterised by a sad, empty, or irritable mood; may be accompanied by cognitive, behavioural, or physiological changes that affect the person’s daily living	[[10], ^b^] [[22], ^d^] [[23], ^a,d^] [[26], ^d^] [27] [[30], ^b^] [[31], ^b^] [[32], ^d^] [[36], ^d^] [[40], ^d^] [[41], ^d^] [[42], ^d^] [[43], ^d^] [[44], ^d^] [[45], ^d^] [[46], ^d^] [[47], ^d^] [[48], ^a^] [52] [[53], ^a^]
Experience of being a lower-limb amputee	Direct and personal knowledge regarding being an amputee	[1,6,17,33,34,37,38,39,50,54]
Prosthesis use	Use of any type of prosthesis by amputees	[[11], ^f^] [13,17] [[22], ^f^] [[27], ^f^] [29,30,31,34,36,37,38,41,49,53,54,55,56] [[57], ^f^] [58,59,60]
QoL/HRQoL	Level of perceived well-being in relation to the socio-cultural context in which individuals live. HRQoL specifically focuses on health aspects	[[11], ^e,g^] [[19], ^e,g^] [[22], ^e^] [27] [[29], ^e^] [[35], ^g^] [[36], ^g^] [[42], ^g^] [[43], ^g^] [[44], ^e^] [46] [[47], ^g^] [[55], ^e^] [[56], ^e^] [[57], ^e^] [[58], ^c^] [[59], ^c^] [[60], ^e^] [61] [[62], ^e^] [[63], ^g^] [64,65] [[66], ^e^] [[67], ^c^] [[68], ^g^]
Social support	Perceived support received from family and friends	[1,11,17,19,22,23,27,29,30,34,37,39,46,52,54,68]

Note. The more frequent questionnaires used in the review were signed in the table: Beck Depression Inventory (BDI) ^a^, Centre for Epidemiologic Studies Depression Scale (CES-D) ^b^, European Quality of Life 5 Dimensions 3 Level Version (EQ-5D-3L) ^c^, Hospital Anxiety and Depression Scale (HADS) ^d^, Short Form Health Survey 36 (SF-36) ^e^, Trinity Amputation and Prosthesis Experiences Scales (TAPES) ^f^, World Health Organisation Quality of Life Brief Version (WHOQOL-BREF) ^g^.

## Data Availability

The data that support the findings of this study are available from the corresponding author, M.M., upon reasonable request.

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
