# Peer review of "What Is Hidden behind Amputation? Quanti-Qualitative Systematic Review on Psychological Adjustment and Quality of Life in Lower Limb Amputees for Non-Traumatic Reasons"

_healthcare, 2023, doi:10.3390/healthcare11111661_

Round 1

Reviewer 1 Report

Dear Authors,

The manuscript is a systematic review that investigates the quality of life, health-related quality of life, and psychological adjustment in non-traumatic lower limb amputees. The review uses the PRISMA statement procedure and analyzes 52 studies retrieved from PubMed, Scopus, and Web of Science databases. The review finds that psychological adjustment, particularly depression with or without anxious symptoms, impacts the QoL/HRQoL in this clinical population. Other factors influencing QoL/HRQoL include subjective characteristics, physical aspects, cause and level of amputation, relational aspects, social support, and physician-patient relationship. The review also finds that emotional-motivational status, depression and/or anxiety symptoms, and acceptance play a crucial role in subsequent rehabilitation processes. The manuscript's conclusion suggests that understanding these issues can provide useful suggestions for clinical and rehabilitative interventions that may be tailored and effective in this clinical population.

However, I can point out a few minor points for improvement:

1.    The search terms used are not described in detail. It would be helpful to provide the specific search strings used for each database searched.

2.    The inclusion criteria do not specify the types of study designs that were included. While the section mentions that studies had to be quantitative or qualitative, it would be helpful to specify the types of study designs that were included in the review.

3.    The section could benefit from a flow diagram illustrating the study selection process as recommended by the PRISMA guidelines. This would provide readers with a visual representation of the study selection process and improve transparency.

4.    It would be helpful to include more information on the specific psychological constructs that were investigated in the included studies and their findings related to QoL/HRQoL and psychological adjustment. This would give the reader a better understanding of the overall results of the review.

The Discussion section of the systematic review provides a good overview of the findings, but there are a few areas that could be improved:

1.    Organization: The discussion could be better organized by grouping related findings together and providing a more cohesive narrative. The current discussion feels disjointed and jumps from one finding to another without much transition.

2.    Comparison with previous literature: The authors could compare their findings with those of previous literature to provide a better context and highlight any discrepancies or agreements.

3.    Limitations: The authors could discuss the limitations of the review and the studies included in it. For example, they could discuss any biases or confounding factors that may have affected the results and suggest areas for future research.

4.    Clinical implications: The authors could discuss the clinical implications of their findings and how they could inform the care and management of patients with lower limb amputations. For example, they could discuss how the findings could inform the development of interventions to improve QoL/HRQoL and psychological adjustment in these patients.

5.    Structure: The text could benefit from clearer section headings and topic sentences to help the reader follow the flow of ideas.

6.    Clarity: Some sentences are long and complex, which may make it difficult for the reader to understand the main point. Simplifying some of the language and breaking up long sentences could improve clarity.

7.    Citation format: The citation format should be consistent throughout the text, and the references should be properly formatted according to the appropriate citation style.

8.    Limitations: While the text acknowledges limitations, it would be helpful to discuss these limitations in more detail and explain how they may impact the findings or conclusions drawn from the reviewed articles.

The conclusions section effectively summarizes the main findings of the review and provides recommendations for future research and clinical practice. However, there are a few ways in which the section could be improved:

1.    The language could be made more concise and clear. For example, instead of saying "it can be argued that QoL in lower limb amputees is initially lower, than it could be improved on the basis of several aspects," it could be phrased more simply as "Quality of life in lower limb amputees is initially lower but can be improved through various factors."

2.    The recommendations for future research and clinical practice could be more specific. For example, the section could suggest particular interventions or areas of study that would be beneficial.

3.    The section could benefit from a stronger concluding statement that ties together the main points of the review and emphasizes the importance of the findings.

The manuscript needs a grammar, spelling and punctuation check that should be done by a native speaker.

Kind regards,

Reviewer 2 Report

The main question addressed by the research is:  What is the current state of literature regarding the quality of life and psychological adjustment in nontraumatic lower limb amputation (LLA) subjects, and what are the risk and protective factors that have not yet been studied in depth?

This topic is relevant in the field as it focuses on understanding the quality of life and psychological adjustment in nontraumatic lower limb amputation (LLA) subjects. While there has been previous research on physical factors influencing amputees, such as age, level of amputation, and presence of comorbidities, this topic seeks to explore the psychological aspects and how they impact quality of life.

The topic addresses a specific gap in the field by aiming to provide a comprehensive overview of the current literature on the psychological well-being of LLA patients. It highlights risk and protective factors that have not been studied in depth, making it an original contribution to the existing knowledge on the subject. Additionally, it acknowledges the methodological challenges in investigating this construct, such as sample heterogeneity and the use of appropriate measurement instruments. By addressing these issues, the research aims to provide a better understanding of the psychological aspects of amputees' lives and their overall well-being.

The conclusions are consistent with the evidence and arguments presented. The authors provide a thorough analysis of the factors affecting QoL/HRQoL among lower limb amputees, emphasizing the importance of a biopsychosocial approach to treatment. The conclusions address the main question posed by providing a summary of the factors influencing QoL/HRQoL and the potential improvements to be made in clinical practice and research.

The authors have followed the PRISMA guidelines for conducting a systematic review and have used a well-defined search strategy. However, they could consider some improvements in their manuscript:

1. Clearer definition of inclusion and exclusion criteria: While the authors have mentioned some inclusion and exclusion criteria, they could provide a clearer and more detailed explanation of these criteria. For instance, they could specify the types of study designs (e.g., cross-sectional, longitudinal) that are eligible for inclusion or exclusion.

2.            Data extraction and synthesis: The authors could provide more information on how they extracted data from the included studies, and how they synthesized the findings.

3. The text could benefit from a more in-depth discussion on potential intervention strategies, including psychological interventions, to improve QoL/HRQoL in lower limb amputees.

4. Terminology: To avoid confusion, it is essential to maintain consistency in the use of terms throughout the text. For example, QoL and HRQoL are used interchangeably, and it might be helpful to clarify if there is a distinction between them or if they refer to the same concept.

Round 2

Reviewer 1 Report

Our new comments in green text.

Dear Authors,

The manuscript is a systematic review that investigates the quality of life, health-related quality of life, and psychological adjustment in non-traumatic lower limb amputees. The review uses the PRISMA statement procedure and analyzes 52 studies retrieved from PubMed, Scopus, and Web of Science databases. The review finds that psychological adjustment, particularly depression with or without anxious symptoms, impacts the QoL/HRQoL in this clinical population. Other factors influencing QoL/HRQoL include subjective characteristics, physical aspects, cause and level of amputation, relational aspects, social support, and physician-patient relationship. The review also finds that emotional-motivational status, depression and/or anxiety symptoms, and acceptance play a crucial role in subsequent rehabilitation processes. The manuscript's conclusion suggests that understanding these issues can provide useful suggestions for clinical and rehabilitative interventions that may be tailored and effective in this clinical population.

However, I can point out a few minor points for improvement:

1.      The search terms used are not described in detail. It would be helpful to provide the specific search strings used for each database searched.

Thank you for this comment which allows us to better explain the search strategy used in the current systematic review. After asking support for data extraction from an external consultant, we adopted the following search string: ("lower extremity amput*" OR "lower limb amput*") AND (psychological OR acceptance OR adherence OR compliance OR "health-related quality of life"). We run the same string in all databases (PubMed, Scopus, Web of Science) and we set the same search limits (query box: title, abstract, keywords, from 2001 to 2021). We properly combined Boolean operators and wildcard characters to focus the search, as well as to detect plural and singular forms of the same terms in all databases. We included synonyms or spelling variations too. As the index terms differ by database, the choice of terms was checked both by clinicians with specific expertise in the amputation field and through the reading of sentinel articles as suggested by Cochrane Handbook for Systematic Reviews of Interventions, 2008.

We have therefore amended in the manuscript the section “search strategy and data extraction” as follows: Three publicly accessible databases (PubMed, Scopus, Web of Science) were used for the electronic literature search using the following terms: ("lower extremity amput*" OR "lower limb amput*") AND (psychological OR acceptance OR adherence OR compliance OR "health-related quality of life"). Articles published from 2001 to 2021 were included in the search. We combined Boolean operators and wildcard characters appropriately to focus the search, and to detect plural and singular forms of the same terms in all databases. We also included synonyms or spelling variations. As the index terms varied between databases, the choice of terms was checked both by clinicians with specific expertise in amputation and by reading of sentinel articles.

            Thank you for your answer to our comment.

2.      The inclusion criteria do not specify the types of study designs that were included. While the section mentions that studies had to be quantitative or qualitative, it would be helpful to specify the types of study designs that were included in the review.

Thank you for this note. We agree with you, and we have therefore amended the manuscript as follows: Articles were considered eligible if they were written in English and published in peer-reviewed journals. Both qualitative and quantitative research was considered, in particular cross-sectional, longitudinal and intervention studies. More specifically, articles were included in the systematic review if they reported the perspective of adult patients with a lower limb amputation, due to a clinical condition (e.g. diabetes, vascular diseases). Publications were included if they discussed amputation in relation to psychological aspects (QoL/HQoL, anxiety, depression, coping...).

            Thank you for your answer to our comment. Nevertheless, was this information added to the manuscript? We didn´t find it there and it would help readers to understand it better.

3.      The section could benefit from a flow diagram illustrating the study selection process as recommended by the PRISMA guidelines. This would provide readers with a visual representation of the study selection process and improve transparency.

We completely agree with you concerning the importance of a flow diagram for transparency. Therefore, Fig. 1 proposes the selection process (i.e. databases, reasons for exclusions, papers retrieved) using the free-downloaded PRISMA diagram. You can see it on page 5. We are more than happy to consider other suggestions you can suggest us to improve this diagram.

            Thank you for your answer to our comment. It seems to be clear enough.

4.      It would be helpful to include more information on the specific psychological constructs that were investigated in the included studies and their findings related to QoL/HRQoL and psychological adjustment. This would give the reader a better understanding of the overall results of the review.

Thank you for your suggestions. We consider to slightly modify the Table 2, adding a column which clarifies better the psychological constructs investigated. We hope that this edited table and the visual map (Fig 2) can allow the reader to catch in a glance the main results and, therefore, help the understanding of details provided in the text. We are confident that this amendment can respond to your requests and ameliorate the manuscript too.

            Thank you for your answer to our comment. Now it is much clearer for readers. The new figure (#2) helps a lot in understanding your analysis.

The Discussion section of the systematic review provides a good overview of the findings, but there are a few areas that could be improved:

Thank you for appreciating the discussion section, we have ameliorated the text as you suggested. Below, you can find our answers:

1.      Organization: The discussion could be better organized by grouping related findings together and providing a more cohesive narrative. The current discussion feels disjointed and jumps from one finding to another without much transition.

Heartfelt thanks for your comment. We re-read and re-organised the entire structure of the discussion. Specifically, we move some parts elsewhere, and we added linkages to promoting cohesiveness of the entire text. We also added section headings and created a new section dedicated to “future directions, strengths and limits”.

Thank you for your answer to our comment. Now it is much clearer for readers, but it would be recommendable to modify this new title as “Future research, strengths and limitations”, which is much usually in research literature.

2.      Comparison with previous literature: The authors could compare their findings with those of previous literature to provide a better context and highlight any discrepancies or agreements.

Thank you for your comment which help us to sharp the discussion. We deeply reviewed and re-organised this section, also adding new references to better compare the current data with previous literature.

We are confident that the manuscript is now improved. However, any other suggestions are more than welcome.

Thank you for your answer to our comment.

3.      Limitations: The authors could discuss the limitations of the review and the studies included in it. For example, they could discuss any biases or confounding factors that may have affected the results and suggest areas for future research.

Thank you for your suggestions. We amended the text, trying to better explain which biases could lead the limitations of this review. Moreover, we re-organised the discussion, creating a specific section dedicated to future directions, strengths and limits.

Thank you for your answer to our comment. We would recommend to follow the answer to our first comment of the Discussion chapter.

4.      Clinical implications: The authors could discuss the clinical implications of their findings and how they could inform the care and management of patients with lower limb amputations. For example, they could discuss how the findings could inform the development of interventions to improve QoL/HRQoL and psychological adjustment in these patients.

Thank you for this note. We tried to ameliorate the clarity creating a section of “future directions, strengths and limits” and providing an example of which kind of interventions seems to be promising for this clinical population. We hope this can better provide clinical implications for daily practice.

Thank you for your answer to our comment. We would recommend to follow the answer to our first comment of the Discussion chapter.

5.      Structure: The text could benefit from clearer section headings and topic sentences to help the reader follow the flow of ideas.

Thank you for your comment which allows us to make clearer the manuscript. We followed your suggestions, and we revised the structure of the manuscript, adding section to help the reader in understanding the discussion.

Thank you for your answer to our comment.

6.      Clarity: Some sentences are long and complex, which may make it difficult for the reader to understand the main point. Simplifying some of the language and breaking up long sentences could improve clarity.

Thank you. We agree with you, and we have conducted a detailed editing of the language in order to detect grammar errors and make the text more easily readable and comprehensible.

Thank you for your answer to our comment. There are still some sentences that need to be improved. Please, re-check all the text again accordingly.

7.      Citation format: The citation format should be consistent throughout the text, and the references should be properly formatted according to the appropriate citation style.

Thank you for this note. We have properly formatted references in order to guarantee consistency.

Thank you for your answer to our comment.

8.    Limitations: While the text acknowledges limitations, it would be helpful to discuss these limitations in more detail and explain how they may impact the findings or conclusions drawn from the reviewed articles.

Thank you for your suggestions. We amended the text, trying to better explain which biases could lead the limitations of this review. Moreover, we re-organised the discussion, creating a specific section dedicated to future directions, strengths and limits.

Thank you for your answer to our comment. We would recommend to follow the answer to our first comment of the Discussion chapter.

The conclusions section effectively summarizes the main findings of the review and provides recommendations for future research and clinical practice. However, there are a few ways in which the section could be improved:

1.      The language could be made more concise and clear. For example, instead of saying "it can be argued that QoL in lower limb amputees is initially lower, than it could be improved on the basis of several aspects," it could be phrased more simply as "Quality of life in lower limb amputees is initially lower but can be improved through various factors."

Thank you for your comment. We have amended the entire text following your suggestions.

Thank you for your answer to our comment.

2.      The recommendations for future research and clinical practice could be more specific. For example, the section could suggest particular interventions or areas of study that would be beneficial.

Thank you for this note. We tried to ameliorate the clarity creating a section of “future directions, strengths and limits” and providing an example of which kind of interventions seems to be promising for this clinical population.

Thank you for your answer to our comment. We would recommend to follow the answer to our first comment of the Discussion chapter.

3.    The section could benefit from a stronger concluding statement that ties together the main points of the review and emphasizes the importance of the findings.

Thank you for your clever suggestions. Accordingly, we conclude the manuscript with the following sentence: Overall, according to the authors, these findings provide useful suggestions not only for research but also clinicians: positive psychological adjustment to the daily challenges can promote better adherence, QoL/HRQoL and medical outcomes in amputee patients.

Thank you for your answer to our comment.

The manuscript needs a grammar, spelling and punctuation check that should be done by a native speaker.

Thank you. We agree with you and we have conducted a detailed editing of the language in order to detect grammar errors and make the text more easily readable and comprehensible.

Finally, authors are encouraged to upload a clean version of the manuscript because all previous corrections make the new version of the manuscript very difficult to read.
